# Subacute Sclerosing Panencephalitis of the Brainstem as a Clinical Entity

**DOI:** 10.3390/medsci5040026

**Published:** 2017-11-07

**Authors:** Pavan S. Upadhyayula, Jason Yang, John K. Yue, Joseph D. Ciacci

**Affiliations:** 1Department of Neurological Surgery, University of California, San Diego, 200 West Arbor Drive #8893, La Jolla, CA 92103, USA; pavan8632@gmail.com (P.S.U.); jasonyangbd@gmail.com (J.Y.); 2Department of Neurological Surgery, University of California, San Francisco, 400 Parnassus Avenue, San Francisco, CA 94122, USA; yuej@neurosurg.ucsf.edu

**Keywords:** brainstem, measles, neurodegeneration, neuroimaging, subacute sclerosing panencephalitis

## Abstract

Subacute sclerosing panencephalitis (SSPE) is a rare progressive neurological disorder of early adolescence caused by persistent infection of the measles virus, which remains prevalent worldwide despite an effective vaccine. SSPE is a devastating disease with a characteristic clinical course in subcortical white matter; however, atypical presentations of brainstem involvement may be seen in rare cases. This review summarizes reports to date on brainstem involvement in SSPE, including the clinical course of disease, neuroimaging presentations, and guidelines for treatment. A comprehensive literature search was performed for English-language publications with keywords “subacute sclerosing panencephalitis” and “brainstem” using the National Library of Medicine PubMed database (March 1981–September 2017). Eleven articles focusing on SSPE of the brainstem were included. Predominant brainstem involvement remains uncharacteristic of SSPE, which may lead to misdiagnosis and poor outcome. A number of case reports have demonstrated brainstem involvement associated with other intracranial lesions commonly presenting in later SSPE stages (III and IV). However, brainstem lesions can appear in all stages, independent of higher cortical structures. The varied clinical presentations complicate diagnosis from a neuroimaging perspective. SSPE of the brainstem is a rare but important clinical entity. It may present like canonical SSPE or with unique clinical features such as absence seizures and pronounced ataxia. While SSPE generally progresses to the brainstem, it can also begin with a primary focus of infection in the brainstem. Awareness of varied SSPE presentations can aid in early diagnosis as well as guide management and treatment.

## 1. Introduction

Measles is a leading cause of morbidity and mortality among young children worldwide, despite the availability of a safe and effective vaccine. During 2015, the United States experienced a multistate measles outbreak; outbreaks such as these are generally related to transmission in under-immunized communities after the disease is introduced from endemic areas [1]. The World Health Organization (WHO) global measles case burden in 2015 was 254,928 cases across all six regions, with an estimated 134,200 deaths [2]. While the WHO reports 189,929 cases in 2016 and 93,943 cases to date in 2017, measles is widely known to be underreported [3]. Although accelerated immunizations have significantly reduced disease burden, measles remains common in developing countries.

The measles virus is a highly contagious enveloped single-stranded negative sense virus, transmitted via the respiratory tract from exposure to infectious aerosols. Initially, the virus demonstrates tropism for CD150+ alveolar macrophages and dendritic cells, leading to systemic spread through lymphoid tissue. Both B- and T-lymphocytes can be infected, with the infection of B-cells creating the pathognomonic syncytial Warthin-Finkeldey cells observed in the upper gastrointestinal (GI) tract [4]. Viral particles migrate to epithelial cells through cell surface receptor nectin-4, leading to the systemic infection in the GI tract, kidney, liver, skin, and in rare cases the central nervous system (CNS) [5]. The rapid dissemination of measles throughout primary, secondary, and tertiary lymphoid tissue creates an acute lymphopenic state soon after infection. This leads to an increased susceptibility to opportunistic infections, specifically in sites that rely on tertiary lymphoid tissues, e.g., GI and pulmonary systems [5].

Measles infection results in a systemic, febrile illness characterized by dry cough, sore throat, muscle aches, coryza, light sensitivity, conjunctivitis, Koplik’s spots inside mouth, and maculopapular rash. Typically, measles is self-limiting; however, measles may result in infrequent complications involving the CNS. Acute disseminated encephalomyelitis and measles inclusion body encephalitis are two acute-onset pathologies that can rapidly progress to coma or death. A third CNS pathology, and the focus of this review, is subacute sclerosing panencephalitis (SSPE) [2].

SSPE is a fatal long-term complication of measles infection, caused by the intracerebral spread of the measles virus leading to neuronal destruction. The incidence of SSPE ranges from 4–11 per 100,000 cases of measles in the U.S. and 23 per 100,000 cases in Israel [6]. Rates among children under one year of age can be up to 18 times higher than children over the age five years, presumably due to immune system immaturity [6]. The latency period between acute measles and the first symptoms of SSPE is usually 4 to 10 years, but can range from one month to 27 years [7]. Most patients survive for one to three years after diagnosis, with a mean survival of about 18 months [8]. Effective treatment remains elusive, and immunization is the only present solution to prevent SSPE [7]. Although canonical SSPE predominantly affects the frontal and temporal lobes, rhombencephalitis due to SSPE affecting the brainstem presents with atypical clinical and imaging findings. The aim of this review is to provide a summary of clinical, neurobiological, semiological, and imaging findings in SSPE with brainstem involvement.

## 2. Methods

The literature search was performed using the National Library of Medicine PubMed database. We included all English-language publications with keywords “subacute sclerosing panencephalitis” and “brainstem” in the title or abstract. This search yielded 15 unique articles. Three study authors (P.S.U., J.K.Y., J.Y.) independently reviewed each article and associated references to determine their relevance to SSPE, indications for brainstem involvement, clinical and neuroimaging presentations, or guidelines for treatment. Two additional articles were included from references. Any discrepancies for determining article inclusion/exclusion were adjudicated by the senior author (J.D.C.). 

Of the 17 articles, six were excluded due to inapplicability to the focus of the current study (one studied non-human primates, two lacked focus on brainstem, three were not accessible). A final total of 11 manuscripts were included in the current review.

## 3. Results

SSPE is thought to progress through four distinct stages. Stage I includes personality changes and behavior disturbance. Stage II shows stereotyped myoclonic muscle contractions, poor coordination, choreoathetosis, tremors, and convulsive motor signs. Stage III is characterized by coma, opisthotonus, decerebrate rigidity, and dystonia. Stage IV progresses to loss of cerebral cortex function, less frequent myoclonus, diminished hypertonia, and eventually death. This presentation is consistent with virus-mediated neuronal destruction of the telencephalon (i.e., occipital, parietal, and frontal lobes) causing personality and cognitive changes in stage I to viral infiltration of the rhombencephalon in the later stages, eventually causing stupor, coma, and death [9,10]. In accordance with these disease stages, a diagnosis of SSPE is based on characteristic clinical signs and symptoms including personality changes, gradual mental deterioration, and myoclonus, coupled with periodic electroencephalogram (EEG) activity and elevated anti-measles immunoglobulin G (IgG) in serum and cerebrospinal fluid (CSF).

Cece et al. describe a study of 76 patients from Turkey with SSPE. Of these, 6.6% received a measles vaccine. Magnetic resonance imaging (MRI) findings most consistently showed signal alteration in the frontal (25.0%), parietal (18.4%), and occipital (15.8%) lobes. Brainstem involvement was only seen in one patient presenting at disease stage II. As such, predominant brainstem involvement is not characteristic of SSPE and can lead to confusion surrounding diagnosis [9]. This progression from cortical structures down to the midbrain is described by Alkan et al. through a case-control study of 18 patients with stage II (*N* = 11) and III (*N* = 7) SSPE and 11 age-matched controls who underwent MRI and diffusion-weighted imaging. MRI of one stage II and five stage III patients revealed brainstem involvement and cerebral atrophy. Neuronal loss, demyelination, and gliosis led to an expansion of extracellular space, as shown by the significant increase in the apparent diffusion coefficient values of the brainstem in patients compared to controls [10]. A case series of 26 SSPE patients by Anlar et al. also showed brainstem involvement in two stage III patients, with associated periventricular white matter and subcortical white matter lesions [11]. Several case reports, however, demonstrated atypical presentations of SSPE with predominant brainstem involvement.

Sharma et al. describe one such case report of a 15-year-old male patient with gradual decline in academic function and new onset generalized tonic-clonic seizures. MRI showed brainstem T2 hyperintensities in the pons and middle cerebellar peduncles. Given the patient’s classic clinical presentation, SSPE was diagnosed and CSF titers confirmed measles titer of 1:512. The authors here stressed the importance of understanding that SSPE can present with brainstem lesions without the involvement of higher cortical structures [12].

Jayakumar et al. describe a series of 15 cases of SSPE, six of whom presented in stage III. Two of these six showed selective atrophy of brainstem on computed tomography (CT) scans with normal cerebral hemispheres. Prior to this case series, brainstem atrophy in late stage disease was thought to be secondary to cerebral atrophy. These CT findings support the idea that imaging pathology is secondary to primary infection by measles virus [13].

The importance of early SSPE diagnosis in brainstem cases is highlighted by Saini et al. in the case of a five-year-old male patient who developed acute onset cerebellar ataxia without focal neurological deficits. MRI showed brainstem lesions involving the pons and cerebellum. Clinical suspicion for an acute demyelinating disease was high and the patient was started on intravenous methylprednisolone at a dose of 30 mg/kg/day. After a brief period of clinical improvement (two to three days), the patient developed myoclonic jerks suspicious for SSPE. This patient had brainstem involvement in stage I of diagnosis. SSPE diagnosis was confirmed on CSF analysis with a measles titer of 1:625 and on EEG that demonstrated stereotyped periodic time-locked complexes consistent with myoclonic jerks [14]. Corticosteroids were avoided due to the risk of complications for a clinically-isolated demyelinating syndrome; in general, steroids have not been beneficial in SSPE and can lead to rapid clinical deterioration [14]. Yaramis and Taşkesen also describe a case of SSPE showing early brainstem involvement characterized by a truncated clinical course and early progression to death. The case was of a 7-year-old boy who presented with severe stage II clinical signs, and his MRI showed lesions in the pons and middle cerebellar peduncles. Hence, early brainstem involvement may designate worsened prognosis in SSPE [8].

The pattern of brainstem involvement in MRI can increase suspicion for SSPE versus other CNS pathologies. Sener describes a case of a 11-year-old presenting with seizures and ataxia nine years after measles infection. Diffusion MRI showed prominent pontine involvement with extension into the midbrain and middle cerebellar peduncle, as well as high signal intensity in the ventral pons. The ventral pons contains corticopontine, corticonuclear, and corticospinal fibers that connect with the cerebral cortex. The dorsal pontine tegmentum, in contrast, is connected to the medulla oblongata [15]. Senol et al. also describe two more cases where T2 MRI showed diffuse involvement of the mesencephalon and pons with sparing of the pontine tegmentum [16].

It is possible for imaging findings to be misleading. Yilmaz et al. describe two cases with early brainstem involvement, the second of which was MRI negative on presentation. This case of a six-year-old female patient was clinically classical. She had frequent myoclonic jerks with EEG, demonstrating periodic high amplitude slow wave bursts. At the six-month follow-up, her MRI showed hyperintense lesions in the pons and cerebellar peduncles [17]. Yaramis and Taşkesen also describe two cases of SSPE showing early brainstem involvement, with one case presenting normal MRI findings upon admission. The case was of a 12-year-old male with clinical symptoms of drop attacks and myoclonic jerks. EEG examination detected 1–3 Hz periodic and bilateral high altitude sharp and slow-wave complexes. The disease progressed rapidly, however, and MRI imaging showed lesions in the brainstem after five months [8].

As described above, EEG findings in SSPE are characteristic and generally include spike-and-wave complexes with 1–3 Hz oscillation. Ishikawa et al. describe a case report of a 10-year-old female patient with a five-month history of absence seizures. Her ictal EEG showed characteristic spike-and-wave complexes followed by 2.5 Hz desynchronization during absence attacks, suggesting lesion origination in the brainstem [18]. This case further describes the varied presentations of SSPE, as seen by absence seizures with mixed SSPE and seizure semiology, from both imaging and EEG perspectives.

## 4. Discussion

Globally, the incidence of measles infections is declining due to aggressive vaccination efforts. In the U.S., measles outbreaks within the last three years can be in part attributed to medical contraindications to vaccinations, travelers from endemic regions outside the U.S., and/or missed vaccination opportunities. Given the potential long-term sequelae of measles infections, effective diagnosis of SSPE is exceedingly important in light of recent events. Our review covers the breadth of literature on the atypical entity of SSPE of the brainstem. By informing clinical judgment and imaging analysis, our goal is to ensure that no cases of SSPE are misdiagnosed, as mortality can range from 50–95% [19,20].

Typically, SSPE progresses through its four stages starting with lesions in the higher cerebral cortices and working its way down to the brainstem. Lesions of the frontal, parietal, and occipital lobes correlate with symptomatology of diminished academic performance, mood dysregulation, and cortical blindness, among others. Although brainstem atrophy is commonly seen as SSPE progresses to stage IV disease, primary lesions noted as MRI hyperintensities at earlier disease stages are rare, with an incidence of roughly 7.6% [11]. Primary brainstem lesions with no associated cortical lesions are even rarer and data is confined to a few case reports [15,16,17]. Primary brainstem lesions are historically suggestive of autoimmune (i.e., autoimmune disseminated encephalomyelitis, multiple sclerosis) or metabolic demyelinating diseases (i.e., osmotic demyelination syndrome) [21]. Importantly, the aforementioned case reports argue against the exclusion of SSPE in the differential diagnosis of primary brainstem lesions. A summary of findings related to SSPE with brainstem lesions, as seen in Table 1, highlights a few important factors. First, although MRI is a valuable diagnostic tool, few patients will have clinical signs and no imaging findings. Diffusion weighted imaging, however, seems to offer greater diagnostic sensitivity with significant differences seen between SSPE patients and controls. When working through a complicated neurological differential, clinical presentation, imaging, and EEG findings are critical.

Notably, all cases of SSPE with pontine involvement preferentially affect the ventral pons and middle cerebellar peduncle [15,16]. The middle cerebellar peduncle is composed of afferents of pontine nuclei that are part of a cortico-ponto-cerebellar axis that transmits the desired positioning of body parts in intended motor movement [22]. It stands to reason that canonical SSPE infection of cortical structures and the predilection of SSPE for the ventral pons may both be due to a similar tropism inherent to the measles virus. It is possible that preferential infection of white matter structures—as seen in cortical descending fibers, basis pontis, and middle cerebellar peduncle—can explain the pattern of SSPE lesion loci. Studies have shown the early involvement of white matter tracts through MRI and diffusion-tensor imaging (DTI) images [23]. Although primary brainstem infections in SSPE are seen as aberrant, they adhere to a pattern of measles virus translocation consistent with more typical presentations.

## 5. Conclusions

SSPE is a rare but serious complication of measles infection. Overall, measles vaccination is effective as first-line prevention against SSPE. Classically, SSPE affects cortical telencephalic structures before progressing to deep brain structures. Here we summarize the literature surrounding atypical presentation of SSPE with primary foci in the brainstem. The varied semiology and imaging presentations are important clinically, as mortality can range from 50–95%. Clinical and pathophysiological understanding of the atypical presentation of this clinical entity and its progression may aid in early diagnosis and targeted treatment, and warrant further research.

## Figures and Tables

**Table 1 medsci-05-00026-t001:** Summary of included studies.

**Case Reports**					
**Author**	**Demographics**	**Presentation**	**EEG**	**Imaging Findings**	**Treatments**
Saini et al. [14]	5-year-old male	Cerebellar ataxia without focal motor deficits or cognitive decline preceded by vesicular, pruritic truncal rash. Anti-measles titers 1:625	Periodic complexes time-locked with myoclonus	Pontine and cerebellar lesions suggestive of demyelination	Methylpredni-solone 30 mg/kg/day for five days
Sharma et al. [12]	15-year-old male	Tonic-clonic seizure with increasing frequency and gradual decline in school performance	Periodic generalized high-amplitude sharp and slow wave of 1–2 s	Hyperintensities in pontine and middle cerebellar regions on T2 and T2 FLAIR, with frontotemporal atrophy	Intrathecal IFN-α, valproate, clonazepam
Yilmaz et al. [17]	9-year-old female and 6-year-old female	Both patients presented with typical SSPE including mental deterioration and myoclonus	Not reported	nine-year-old: 2 cm × 2.5 cm focal pontine lesion on T2 MRI; six-year-old: hyperintense pontine and cerebellar peduncular lesions on T2 MRI	Not reported
Sener [15]	11-year-old male	Not reported	Not reported	DWI showing prominent pontine involvement with sparing of tegmentum and longitudinal nerve bundles	Not reported
Ishikawa et al. [18]	10-year-old female	Atypical absence seizures for three months	Diffuse spike and wave complexes of 2.5 Hz with desynchronization during absence attacks	Not reported	Not reported
**Case Series and Reviews**					
**Author**	**Description**	**Results**	**Imaging Findings**		
Alkan et al. [10]	18 SSPE patients 10 controls with MRI/DWI	33% of SSPE patients had brainstem involvement; greater proportion of Stage III SSPE had brainstem involvement compared to Stage II (71.4% vs. 9.0%); increased ADC values were associated with disease stage	ADC values in SSPE patients across seven regions (frontal, parietooccipital, cerebellar and deep white matter, basal ganglia, thalamus, and brainstem) were significantly increased from controls, *p* < 0.05		
Jayakumar et al. [13]	15 SSPE patients with CT	Brainstem involvement in two of 15 patients; one patient with Stage III disease had normal CT	Stage II: cerebral edema, diffuse white matter attenuation. Stage III/IV: cerebral, cerebellar and brainstem atrophy, white matter hypodensities		
Anlar et al. [11]	26 SSPE patients with T2 and T2 FLAIR MRI	Lesions originated in cortical or subcortical white matter; extent of white matter lesions was not correlated with neurologic status; three patients had normal scans	T2-hyperintense lesions progressed from subcortical white matter to periventricular white matter; basal ganglia and brainstem lesions were rare		
Cece et al. [9]	76 SSPE patients with MRI	Stage I/II *n* = 48, stage III *n* = 25, stage IV *n* = 3; significant subset of presented with myoclonic seizures (57.9%) and/or behavior alteration (30.3%)	MRI normal in 21 patients (Stage I/II *n* = 19); three patients had brainstem involvement (Stage II *n* = 1, Stage IV *n* = 2); T2-hyperintense lesions were predominantly in the periventricular and subcortical white matter		

EEG: electroencephalogram; SSPE: sub-acute sclerosing pan-encephalitis; FLAIR: fluid attenuated inversion recovery; IFN-α: interferon-alpha; MRI: magnetic resonance imaging; DWI: diffusion weighted image; ADC: apparent diffusion coefficient; CT: computed tomography.

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
