# Peer review of "Subacute Sclerosing Panencephalitis of the Brainstem as a Clinical Entity"

_medsci, 2017, doi:10.3390/medsci5040026_

Round 1

Reviewer 1 Report

The article addresses an interesting albeit rare presentation of SSPE. The article is very well written and of interest to readers in the field of neuroimmunology and viral CNS infection. The main results are based on a PubMed literature research and not on an actually encountered case.

Author Response

We greatly appreciate the reviewers comments and feedback, and thank him/her for their time in reading our manuscript.

Reviewer 2 Report

In this article, Yang et al. reviewed published literature on a rare presentation of SSPE, a severe neurological complication of measles virus infection. Specifically the authors review SSPE cases presenting with early brainstem involvement. Understanding the different presentations of SSPE is important because of the high mortality associated with SSPE, and because of recent reports stating that SSPE is considerably underdiagnosed, highlighting that the true burden of SSPE is unknown. Because the risk for and severity of SSPE is greater among infants, more studies on this devastating complication are important to emphasize the importance of lowering the age of vaccination to 6 months in outbreak settings. This is well-written and well-researched. I only have minor comments and a major comment. The major comment may the beyond the scope of this paper so it is only for consideration.

Minor:

-Introduction: There was a measles outbreak early in 2015, although the source of the outbreak was unknown, and it was not the largest since 2000. Could replace the sentence stating that the US reported the most cases since elimination in 2014, and that measles outbreaks are related to transmission in underimmunized communities after disease introduction from endemic areas.

-Methods: May be important to mention here or in the introduction what are the criteria to diagnose SSPE. Although understanding the imaging of SSPE is important, I believe imaging is not a criteria.

-Results: In the discussion of the Yilmaz et al. report, it is unclear if the 6 months is after development of symptoms or at 6 months of age.

-Discussion: Would rephrase sentence about outbreaks occurring in immigrant and anti-vaccination communities. Outbreaks occur in the US because measles remains endemic in many areas of the world and unvaccinated returning travelers to the US introduce measles into pockets of under-immunized individuals. Lack of vaccination can be related to an anti-vaccination sentiment, but many people who refuse vaccine accept vaccination during outbreaks, and there are many other reasons such as missed opportunities to vaccination, and age (infants) and medical (immunocompromised) contraindications to vaccination.

-Conclusion: Would rephrase “it is not universally effective”. The vaccine is highly efficacious and safe; it is among the best of the available vaccines.

Major:

-From the epidemiological standpoint, the most interesting aspect of this review is the suggestion that early brainstem involvement may be a marker for higher severity, and thus perhaps brainstem involvement may also be more common when measles occurs at a younger age. I think a simple table listing the demographic, clinical, laboratory, EEG, and radiologic characteristics of SSPE cases with brainstem involvement identified in this review would be of benefit. In addition, expanding on the discussion about the possible differences on severity and age based on brainstem versus cortical involvement may be of interest.

Author Response

We thank the Reviewer for his/her comments and support. We agree with the critiques and have provided our responses below. The Reviewer’s comments are reposted in black, and our responses are posted in red.

-Introduction: There was a measles outbreak early in 2015, although the source of the outbreak was unknown, and it was not the largest since 2000. Could replace the sentence stating that the US reported the most cases since elimination in 2014, and that measles outbreaks are related to transmission in under-immunized communities after disease introduction from endemic areas.

We agree with the Reviewer’s accurate description and have revised the Introduction to read:

"During 2015, the United States (U.S.) experienced a multistate measles outbreak; outbreaks such as these are generally related to transmission in underimmunized communities after disease is introduced from endemic areas [1]."

-Methods: May be important to mention here or in the introduction what are the criteria to diagnose SSPE. Although understanding the imaging of SSPE is important, I believe imaging is not a criteria.

We wholeheartedly agree. Rather than include this in the Methods section, we have added to the first section of the Results to read:

"In accordance with these disease stages, a diagnosis of SSPE is based on characteristic clinical signs and symptoms including personality changes, gradual mental deterioration, and myoclonus, coupled with periodic EEG activity and elevated anti-measles IgG in serum and CSF."

Results: In the discussion of the Yilmaz et al. report, it is unclear if the 6 months is after development of symptoms or at 6 months of age.

We agree that this statement is ambiguous. We have rephrased it to read:

"At six month follow-up, her MRI showed hyperintense lesions in the pons and cerebellar peduncles."

Discussion: Would rephrase sentence about outbreaks occurring in immigrant and anti-vaccination communities. Outbreaks occur in the US because measles remains endemic in many areas of the world and unvaccinated returning travelers to the US introduce measles into pockets of under-immunized individuals. Lack of vaccination can be related to an anti-vaccination sentiment, but many people who refuse vaccine accept vaccination during outbreaks, and there are many other reasons such as missed opportunities to vaccination, and age (infants) and medical (immunocompromised) contraindications to vaccination.

We agree that the language could use revision. This section now reads:

"In the U.S., measles outbreaks within the last three years can be in part attributed to medical contraindications to vaccinations, travelers from endemic regions outside the U.S., and/or missed vaccination opportunities."

Conclusion: Would rephrase “it is not universally effective”. The vaccine is highly efficacious and safe; it is among the best of the available vaccines.

We agree that the original statement could be misconstrued. We have rephrased it to state:

"Overall, measles vaccination is effective as first line prevention against SSPE."

Major:

-From the epidemiological standpoint, the most interesting aspect of this review is the suggestion that early brainstem involvement may be a marker for higher severity, and thus perhaps brainstem involvement may also be more common when measles occurs at a younger age. I think a simple table listing the demographic, clinical, laboratory, EEG, and radiologic characteristics of SSPE cases with brainstem involvement identified in this review would be of benefit. In addition, expanding on the discussion about the possible differences on severity and age based on brainstem versus cortical involvement may be of interest.

We think this is an excellent idea and have included this table, with relevant changes to the Discussion, in the updated version of our manuscript. The changes we made are:

“A summary of findings related to SSPE with brainstem lesions, as seen in Table 1, highlights a few important factors. First, although MRI is a valuable diagnostic tool, few patients will have clinical signs and no imaging findings. Diffusion weighted imaging, however, seems to offer greater diagnostic sensitivity with significant differences seen between SSPE patients and controls. When working through a complicated neurological differential, clinical presentation, imaging and EEG findings are critical.”